# Research on Digital Forensics Analyzing Heterogeneous Internet of Things Incident Investigations

**Dong-Hyuk Shin** [1], **Seung-Ju Han** [1], **Yu-Bin Kim** [1] **and Ieck-Chae Euom** [2,*]

[1] System Security Research Center, Chonnam National University, Gwangju 61186, Republic of Korea; shindh2@jnu.ac.kr (D.-H.S.); sjhan@jnu.ac.kr (S.-J.H.); kingyoubin97@jnu.ac.kr (Y.-B.K.)
[2] Department of Data Science, Chonnam National University, Gwangju 61186, Republic of Korea
* Correspondence: iceuom@jnu.ac.kr

**Abstract:** In the landscape of the Fourth Industrial Revolution, the integration of the Internet of Things (IoT) in smart-home technology presents intricate challenges for digital forensics. This study investigates these challenges, focusing on developing forensic methodologies suitable for the diverse and complex world of smart-home IoT devices. This research is contextualized within the rising trend of interconnected smart homes and their associated cybersecurity vulnerabilities. Methodologically, we formulate a comprehensive approach combining open-source intelligence, application, network, and hardware analyses, aiming to accommodate the operational and data storage characteristics of various IoT devices. Extensive experiments were conducted on prevalent platforms, such as Samsung SmartThings, Aqara, QNAP NAS, and Hikvision IP cameras, to validate the proposed methodology. These experiments revealed crucial insights into the complexities of forensic data acquisition in smart-home environments, emphasizing the need for customized forensic strategies tailored to the specific attributes of various IoT devices. The study significantly advances the field of IoT digital forensics and provides a foundational framework for future explorations into broader IoT scenarios. It underscores the need for evolving forensic methodologies to keep pace with rapid technological advancements in IoT.

**Keywords:** Internet of Things; incident investigation; smart home; digital forensics





## 1. Introduction

The Internet of Things (IoT) is a cornerstone of the Fourth Industrial Revolution, deeply integrated into daily life and offering a wide range of services, especially in smart-home networks. The IoT landscape is expanding rapidly, with projections indicating that the number of IoT devices will surpass 30 billion by 2025, impacting various sectors including home automation, healthcare, and manufacturing [1]. This expansion is particularly evident in the smart-home market, which is expected to reach a value of USD 174 billion by 2025 and continue to grow to USD 231.6 billion by 2028, reflecting an approximate annual growth rate of 10% [2,3].

As IoT technology expands, it confronts critical challenges, notably in power consumption and cost, which impede the implementation of robust security measures and amplify vulnerabilities [4–6]. Additionally, the diverse range of hardware and operating systems across IoT devices introduces significant obstacles in data collection and analysis during cybersecurity incidents. This heterogeneity not only complicates cyber incident response, but also intensifies the complexity of forensic investigations, posing challenges to conducting effective and precise responses to these incidents [7].

In recent studies on forensic analysis of IoT devices, enhanced methodologies for data collection and forensic analysis have been proposed, tailored to the heterogeneous nature of IoT systems in the event of IoT-related incidents [8]. These advancements in research underscore the evolving strategies to address IoT security issues, providing a backdrop

for our current study. Our objective is to refine these approaches by establishing effective data collection and analysis methods that adapt to various information sources, thereby enabling prompt and accurate responses to cybersecurity incidents. This, in turn, aims to strengthen the security of IoT devices and mitigate the impact of such incidents. This study proposes data collection and analysis techniques optimized for investigating intrusion incidents targeting IoT devices, thereby contributing to the reinforcement of the security framework for IoT devices.

The structure of this paper is organized as follows. Section 1 describes the current state and importance of IoT technology, focusing on smart homes. Next, Section 2 examines the challenges faced by the current IoT technology and digital forensics and discusses the future direction of this research through a comparison of related literature. Then, Section 3 briefly explains the objectives and methodology of this research. Section 4 describes the artifacts obtained by applying the proposed data collection and analysis techniques to actual IoT devices. Further, Section 5 sets up the hypothetical scenarios and identifies valid artifacts for incident investigation by applying the proposed data acquisition methodology. Section 6 details the results of such research. The final section presents several conclusions based on the research results and discussion.

## 2. Related Work

In this section, we meticulously survey the existing literature, shedding light on the advancements and challenges in the realm of IoT digital forensics, with a particular emphasis on smart-home environments. This critical review not only contextualizes our research within the broader scholarly discourse, but also identifies gaps and opportunities for innovation in forensic methodologies tailored to diverse IoT scenarios.

### 2.1. Internet of Things Services and Threat

The integration of IoT into modern technological paradigms entails a symbiosis of varied components, including services, platforms, networks, and devices, collectively enhancing the functionality of daily life systems [9]. These services, encompassing applications from personal smart-home systems to industrial manufacturing processes, are underpinned by platforms that harness advanced technological capabilities like artificial intelligence (AI) and big data [10,11]. Such platforms are instrumental in processing and managing the extensive data flow emanating from IoT devices, thereby ensuring their optimized operation. Networks, incorporating both wireless and wired communication modalities, constitute the fundamental infrastructure for uninterrupted device connectivity. At the heart of these IoT applications lie the devices themselves, outfitted with sensors and actuators, and ranging broadly from personal wearable health monitors to industrial automated machinery [12–14].

The realm of IoT services can be divided into private, public, and industrial segments, each addressing distinct needs and presenting unique challenges. Private sector services primarily augment personal convenience and lifestyle quality, whereas public sector services, often government-led, address broader societal concerns such as public safety and environmental surveillance. Conversely, industrial sector services are business-oriented, focusing on augmenting efficiency and a competitive edge [15]. These diverse applications, as outlined in Table 1, highlight the expansive nature of IoT and emphasize the need for customized strategies in both technological advancement and cybersecurity.

In the domain of IoT, threats can be categorized into three primary types: software-based, network-based, and hardware-based threats. Each category represents a unique facet of IoT vulnerabilities, and these often interlink, leading to amplified risks and more complex attack scenarios [16].

**Table 1.** Classification of Internet of Things services and their applications and technology elements.

| Sector | Service Type | Application | Technology |
|---|---|---|---|
| Private | Home<br>Health care<br>Agriculture | Home automation<br>Remote patient monitoring<br>Precision farming | Smart devices<br>Wearable devices<br>Sensor networks |
| Public | Safety<br>Environmental<br>Energy<br>Transportation | Smart city<br>Pollution monitoring<br>Energy management<br>Connected car | Emergency response<br>Data analytics<br>Smart grids<br>V2X communication |
| Industrial | Manufacturing<br>Defense | Predictive maintenance<br>Mobile wearable networks | Sensors<br>Secure networks |

Note: V2X: vehicle to everything.

Software-based threats in IoT predominantly target the application layer of these systems. These attacks bear resemblance to traditional attacks in standard IT environments and typically involve well-established types of cyber attacks, often executed through automated tools. The consequences of such attacks can be severe, including unauthorized access to critical data and control over IoT devices and services.

Network-based threats aim at exploiting weaknesses within the IoT system's network infrastructure. IoT devices use a mix of wired and wireless communication technologies. Attacks in this category can disrupt or intercept communication channels, potentially leading to unauthorized data access or alteration of data during transmission.

Hardware-based threats leverage security lapses present in the physical components of IoT systems. These attacks necessitate physical access and a sophisticated understanding of the hardware's design and components. They can expose sensitive information not accessible via software-based methods and can alter the operation of connected devices. This includes the theft of encryption keys and taking control over system operations.

*2.2. Challenges of Internet of Things Digital Forensics*

Regarding IoT services and their forensic implications, the challenges arising from the diversity of devices and the intricacy of data retrieval are of the utmost importance. Figure 1 provides an illustrative framework of a smart-home ecosystem, encompassing the multitiered IoT architecture. The framework commences with sensor devices that gather environmental data, progressing through the network layer, which employs protocols (e.g., Zigbee, Bluetooth, Z-Wave, and Wi-Fi) to establish broader connectivity [17,18].

The diagram culminates at the smart hub, operating as the central node facilitating communication between physical devices and the user-interface application. Furthermore, the hub extends into the cloud to enhance data processing and storage capabilities. This layered architecture mirrors the operational dynamics of IoT systems, underscoring the potential forensic challenges that arise due to the decentralized and diverse nature of IoT environments. These challenges are explored in the subsequent sections of this research. Consequently, the schematic in Figure 1 is crucial for comprehending the interconnected IoT landscape and the ensuing digital forensic hurdles within smart-home networks.

The process of obtaining data from IoT devices presents distinctive challenges, due to the limited resources and diverse operational environments of these devices [19]. Often, IoT devices use different proprietary data storage mechanisms as a result of their constrained resources. This diversity, along with limitations in memory, battery life, and network bandwidth, makes data access and retrieval more complicated [20]. Moreover, IoT devices typically have limited memory and processing capabilities, despite having various sensors and functionalities. This limitation becomes problematic as these systems continuously operate and generate substantial volumes of data from numerous sensors, leading to rapid data overwriting. Transferring data to external storage devices as a workaround is hindered by the risk of data integrity loss during transmission.

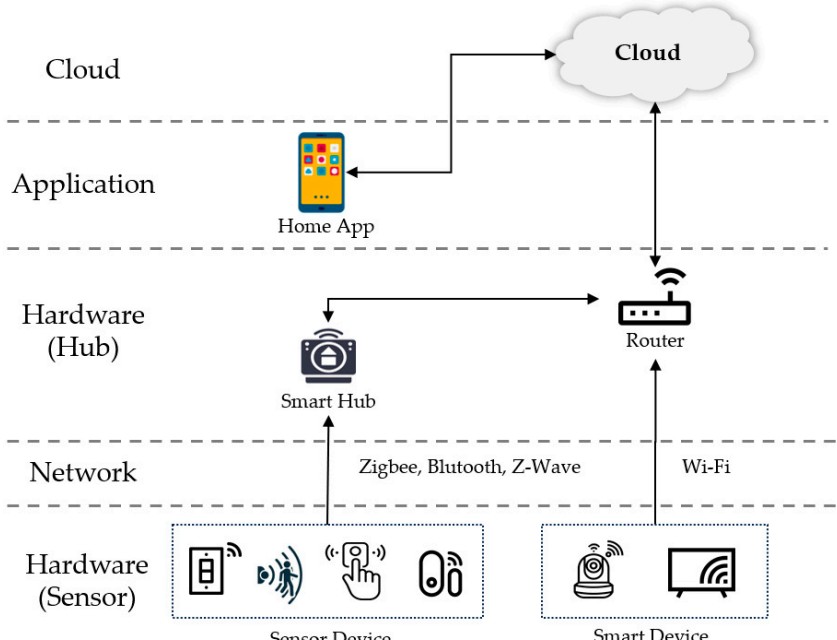

**Figure 1.** Smart-home infrastructure.

The heterogeneity of IoT devices encompasses a wide range of products, including web cameras, digital locks, routers, and thermostats, and adds another layer of complexity. The variety of data formats and types of evidence necessitates unique methodologies for collecting and analyzing data for each device type [21]. Specialized tools and techniques for extracting and interpreting data from these diverse systems have been developed. However, questions regarding their reliability and the potential risk of compromising data integrity during the collection process persist.

Last, the distributed nature of IoT data across multiple platforms, extending beyond the devices to cloud network areas, introduces another level of complexity. The data stored in the cloud and their restricted usage within the network and IoT devices present formidable challenges in integrating data across these distributed ecosystems [22,23].

In conclusion, the challenges associated with acquiring data from IoT devices encompass a range of problems, including limited resources, insufficient processing capabilities, encryption complexities, device heterogeneity, and the distributed nature of IoT data. Addressing these challenges is vital for effective forensic analysis and requires innovative approaches in data collection and analysis methodologies [24].

### 2.3. Literature Review on Smart-Home Digital Forensics

Building on the insights from the previous sections, the field of smart-home digital forensics has experienced significant advancements, primarily driven by the need to address the escalating cybersecurity challenges in IoT environments. Research, such as that by Kang et al. [25], has explored forensic analysis specific to Xiaomi smart-home devices, offering a methodology for extracting and analyzing artifacts crucial for forensic investigations. This work underscores the importance of tailored forensic approaches for different IoT ecosystems.

Complementing this research, Plachkinova et al. [26] reviewed the current literature, identifying five key research trends in smart-home security, privacy, and digital forensics. Their research plays a vital role in contextualizing the broader landscape of smart-home digital forensics and underscores the complexities of security breaches and privacy violations in these environments.

Similarly, Hariyadi et al. [27] proposed a forensic investigation methodology for smart routers within smart-home networks using standardized frameworks, such as

NIST SP 800-86 and SNI ISO/IEC 27037:2014. This approach exemplifies the structured and methodical process required for effective IoT device forensics. Kim and Shon [28] extended the scope of digital forensics to E-IoT devices within smart cities, focusing on identifying vulnerabilities and proposing novel methodologies for data acquisition and analysis.

Moreover, Kim et al. [29] discussed the diverse nature of data acquisition, classification, and analysis from various smart-home devices. Their work is notable for its practical approach to addressing the multifaceted nature of smart-home data for forensic purposes. Additionally, Kaushik et al. [30] provided a comprehensive overview of the advancements and ongoing challenges in IoT forensics, noting potential future directions in this field.

Preda [31] examined the challenges in digital forensics of IoT environments, with a specific focus on smart heating systems and providing empirical methods for investigating security incidents in IoT systems, adding to the repertoire of forensic methodologies. Finally, Awasthi et al. [32] presented an in-depth forensic analysis of the Almond smart-home hub, offering detailed methodologies for data acquisition and analysis. These methodologies are critical for understanding user interactions and data management in IoT devices.

These collective studies underscore the dynamic and evolving nature of digital forensics in smart-home environments. This research highlights the need for continuous innovation in forensic techniques and methodologies to address the emerging technologies and threats effectively in these interconnected systems, see Table 2.

**Table 2.** Comparative analysis of related work in smart-home digital forensics.

| Related Work | Artifact Acquisition | | | | Incident Investigation |
|---|---|---|---|---|---|
| | Cloud | Application | Hardware | Network | |
| Kang et al., (2021) [25] | O | O | O | X | O |
| Plachkinova et al., (2016) [26] | X | O | O | X | X |
| Hariyadi et al., (2023) [27] | X | O | O | O | O |
| Kim and Shon (2023) [28] | O | X | O | O | O |
| Kim et al., (2020) [29] | O | O | X | O | O |
| Kaushik et al., (2023) [30] | O | O | X | X | X |
| Preda (2020) [31] | O | X | X | O | O |
| Awasthi et al., (2018) [32] | X | O | O | O | O |

## 3. Proposed Methodology

This section proposes a comprehensive methodology for investigating hacking incidents in smart-home platforms, encompassing four main aspects: open-source intelligence (OSINT), network, application, and hardware. This classification reflects the complexity and diversity of smart-home environments and considers the characteristics of heterogeneous IoT devices to enable holistic data collection and analysis. The methodology proposed in Figure 2 builds on existing forensic approaches in IoT, as discussed by Hutchinson et al. [33]. The key distinction of the proposed methodology is its focus on consistent data acquisition from an incident investigation perspective in already implemented smart-home environments.

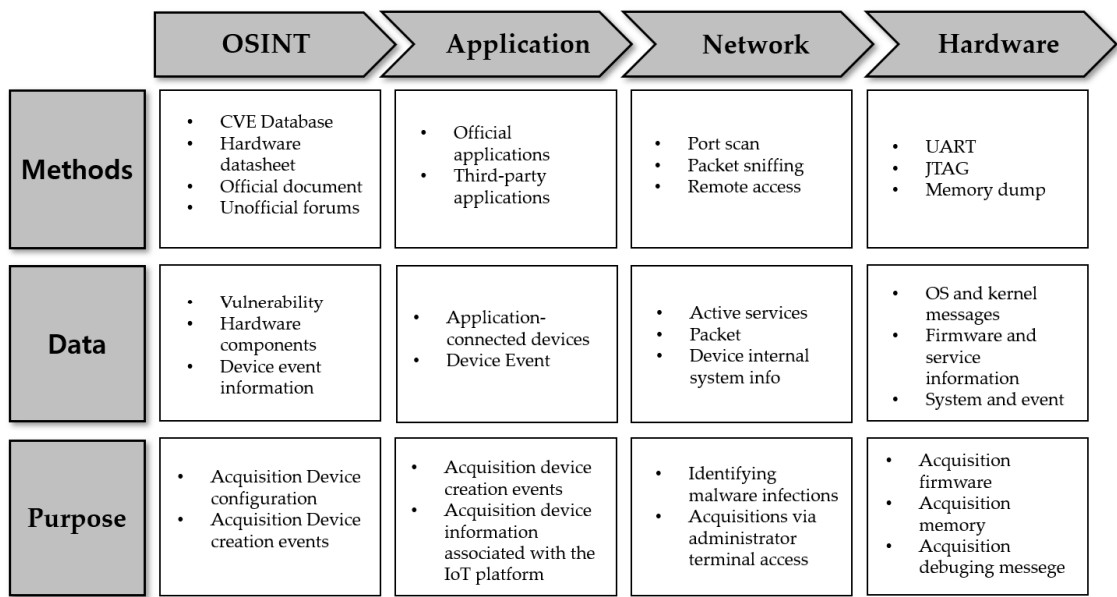

**Figure 2.** Methodology for smart-home data acquisition.

*3.1. OSINT-Based Incident Investigation Process*

The investigation of hacking incidents from the perspective of OSINT is a preparatory phase for the commencement of comprehensive information gathering. The OSINT-based process is crucial for assessing the security status of IoT devices and identifying potential security threats, providing essential information to identify and analyze security threats in the IoT environment. The process of collecting and analyzing OSINT data gathers and analyzes a wide range of information from various online resources and IoT devices. OSINT-based information collection is valuable because it allows for the acquisition of a wide range of information at a low cost. However, despite the significant advantages in the diversity and accessibility of information, professional judgment and analysis are required in terms of the reliability and interpretation of the information. Selecting relevant and accurate information from the vast volume of available data can be challenging. The interpretation of information can vary depending on the analyst's expertise and experience, which can hinder information collection. The investigation from the perspective of OSINT involves collecting the following information:

● Known vulnerabilities and exploits;
● Known components of the device;
● Accessible application programming interfaces (APIs);
● Tools for investigating hacking incidents.

Collecting information about APIs can provide access to data from typically hard-to-reach repositories, such as the cloud. APIs can be categorized as public or private. Public APIs can be easily researched through developer-friendly sources, such as official documentation, without significant costs. However, some APIs are not publicly disclosed for security reasons. In such cases, the nature and type of APIs can be inferred by analyzing open-source tools that control smart-home platforms.

Moreover, APIs are categorized from a forensic perspective into three categories: essential, considerable, and irrelevant. The essential category includes essential information for forensic investigation, such as core data (e.g., the device log information). The considerable category refers to information related to the device or user that is insufficient for immediate use in an investigation. Finally, irrelevant information pertains to details irrelevant to the incident. This categorization helps effectively analyze data, focusing on important information during the investigation.

Ultimately, this information collection provides insights into the pathways through which attackers have caused security breaches. Therefore, it is crucial to collect information that could be systematically exploited. Figure 3 presents a flowchart depicting the method of investigating hacking incidents from an OSINT perspective.

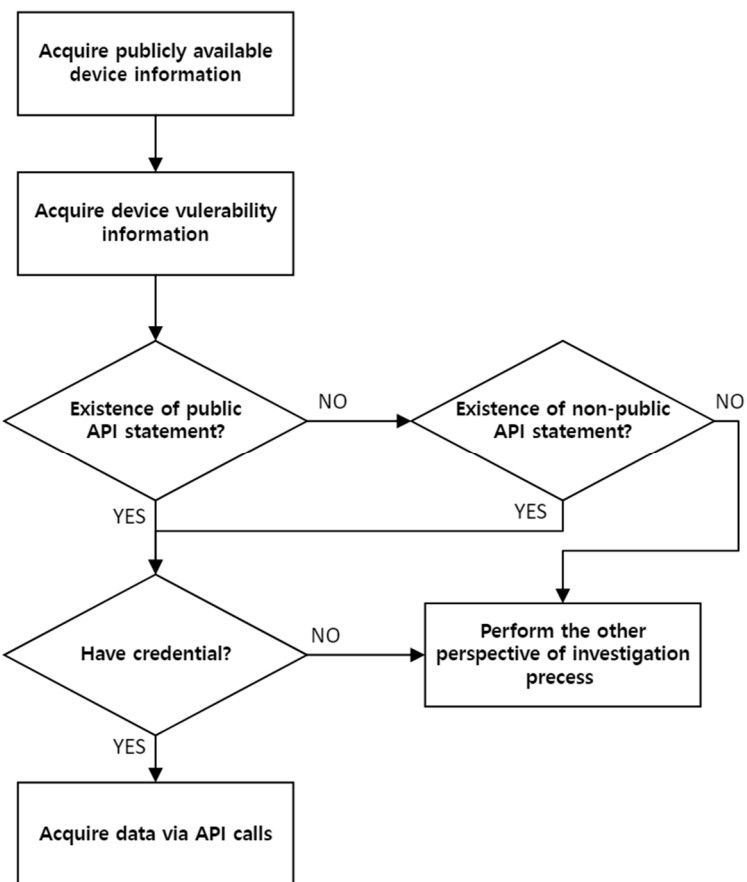

**Figure 3.** OSINT-based data acquisition process.

*3.2. Application-Based Incident Investigation Process*

Investigating hacking incidents from the application perspective targets the mobile applications that control smart-home platforms and the mobile devices on which these applications are installed. This investigation should follow an OSINT-based investigation process. Mobile applications that are connected to various IoT devices allow users to access diverse functionalities and information, most of which are stored in the private local storage of the mobile device. A process called "rooting", which grants privileged access, is essential to access this storage.

One of the critical advantages of this internal storage access approach is that it does not require additional account information or separate access permissions. Unlike typical API calls, rooting allows direct access to the application's internal storage, which is very useful in hacking incident investigations. This internal storage contains various data (e.g., DB files and XML files), including account information used in the app and the device log information, which are crucial for hacking incident investigations. The investigation from the perspective of the application involves collecting the following information:

- Device event logs;
- Network identifiers (e.g., medium access control (MAC) addresses);
- User customization information.

The data from the internal storage can be vast, so they are categorized from a forensic perspective into three categories in the same way as APIs: essential, considerable, and

irrelevant. Figure 4 presents a flowchart of the methods of incident investigation from the application perspective.

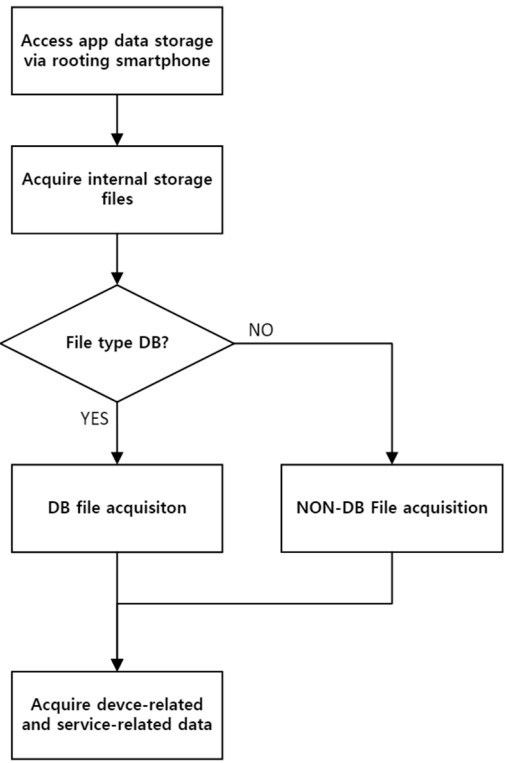

**Figure 4.** Application-based data acquisition process.

### 3.3. Network-Based Incident Investigation Process

If the device remains active after the application-side incident investigation, a network-side incident investigation is conducted. This investigation involves obtaining information from the activated device, allowing for the collection of volatile data. In addition, as IoT devices often communicate within internal networks using routers, access to the internal network is crucial for a thorough network investigation. However, in cases where low-power protocols like smart sensors are used for communication, packet collection can be achieved through sniffing tools without accessing the internal network. The investigation from the network perspective involves collecting the following information:

- Network packets;
- Active services;
- Device system information through remote access.

This information can be collected when the device is active. It is critical for identifying and assessing security breaches.

Most IoT devices have encrypted communications, making it challenging to collect data from packets. However, even if the packet content is encrypted, network traffic collection through sniffing provides crucial data beyond the packet content. For example, metadata, such as internet protocol (IP) addresses, port numbers, and packet characteristics, are indispensable for understanding activities and traffic flow within the network. IP addresses and port numbers identify specific network devices and services, and analyzing this information can determine which devices are active on the network and the services with which they are communicating. Additionally, analyzing the packet characteristics, such as the size, frequency, and traffic pattern, can detect abnormal device activities or potential security threats. For instance, unusually sizable data transfers or consistent traffic to a specific destination can indicate that a device is infected with malware.

Furthermore, if remote access services (e.g., telnet) are available or if management services exist, they can be identified through port scanning to obtain additional information. Remote access services allow administrators or investigators to directly access a device over the network and inspect system settings, log files, and running processes. The information available through management ports varies depending on the device and is primarily used to acquire device logs or system setting information. Access through these ports can provide detailed operating status, system logs, and configuration settings of each device, allowing for a deeper understanding of the system information. Figure 5 presents a flowchart of the network aspect of incident investigation methods.

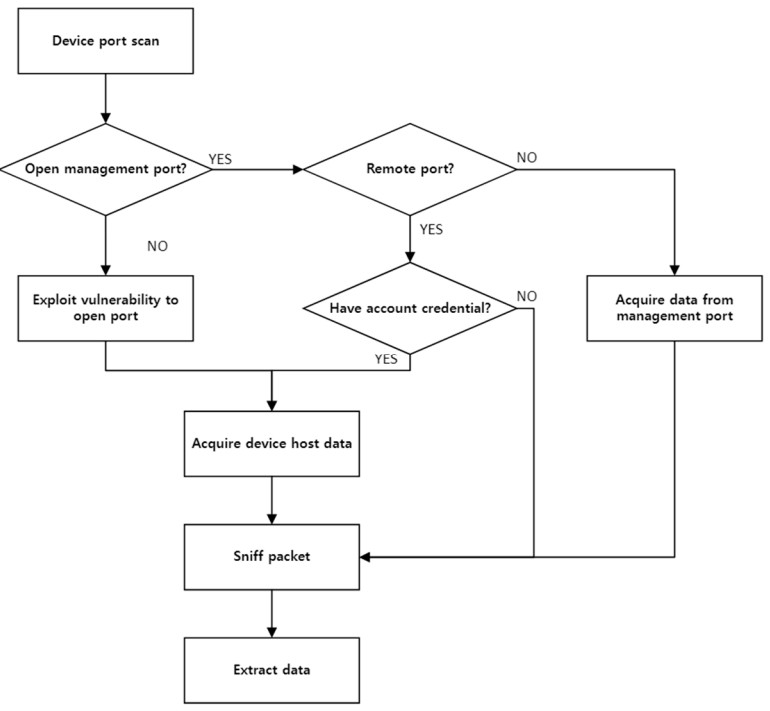

**Figure 5.** Network-based data acquisition process flow.

### 3.4. Hardware-Based Incident Investigation Process

The process of hardware-based data collection and analysis is crucial for understanding the operational mechanics and security posture of IoT devices, as well as identifying potential security vulnerabilities. This process involves collecting a wide range of information, such as operating system (OS) and kernel messages, firmware details, service information, system and event logs, and operational data. From a forensic perspective, these data are essential for dissecting and understanding the intricacies of IoT environments, especially in the context of security incidents. When a device is in an inactive state or fails to acquire meaningful data from previous processes, an investigation can be conducted from a hardware perspective. Further, IoT devices often have debugging interfaces for management and control purposes. Users can read messages or access the management terminal using appropriate tools for each interface. Furthermore, information can be obtained by removing components, such as the flash memory, from the device and reading the memory. The objectives of conducting a hardware-level investigation include the following:

- Accessing internal storage;
- Performing memory dumps;
- Acquiring firmware.

Figure 6 presents a flowchart of the hardware aspect of incident investigation methods, which can appear complex due to the various ways data can be collected compared with other procedures. The investigation methods can be broadly divided into two types: chip-

off and interface methods. These incident investigation methods can be a powerful source of information, but caution is required because they can lead to permanent damage to IoT devices. Additionally, if the data collection methods specific to each component are not followed, there can be problems in interpreting the information.

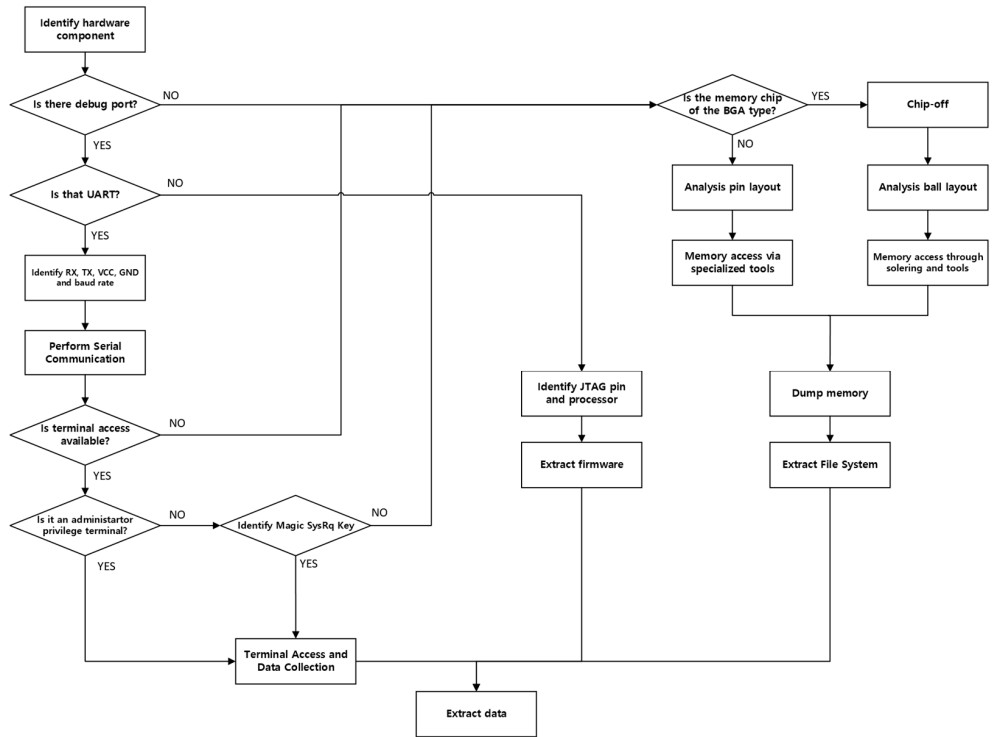

**Figure 6.** Hardware-based data acquisition process flow.

## 4. Experiments

This section describes the experimental setup designed to collect data from four aspects of a smart-home environment, as illustrated in Figure 7. The smart-home testbed used Samsung SmartThings and Aqara platforms, incorporating the following devices: a smart-home hub, smart cameras, smart sensors and storage, and an IP camera.

The choice of the SmartThings and Aqara platforms, known for their extensive user base and compatibility with a wide range of IoT devices, enhances the representativeness of the experiment. These platforms facilitate a systematic analysis of the characteristics and vulnerabilities of security breaches that can occur in cloud-based IoT environments. Both platforms connect to cloud servers through a router.

Conversely, the Hikvision IP cameras and QNAP network-attached storage (NAS) were selected due to their independent network connections, which are not reliant on cloud services. These devices are ideal for investigating the characteristics and vulnerabilities of security breaches in cloud-independent IoT environments with local data storage. Notably, the Hikvision cameras are directly connected to NAS through a router.

For the experiment, we configured a network environment where all devices were connected to the same router, which was linked to an external network. This setup allowed for the inclusion of cloud-based and cloud-independent IoT environments, facilitating the replication and analysis of various security breach scenarios that might occur in real-world settings. Table 3 presents detailed information regarding the devices used in this experiment.

Building on the experimental environment outlined in Table 3, Table 4 further details the methodology for the smart-home application analysis and data environment. This table includes the tools and software used for monitoring, controlling, and analyzing the data from smart-home devices.

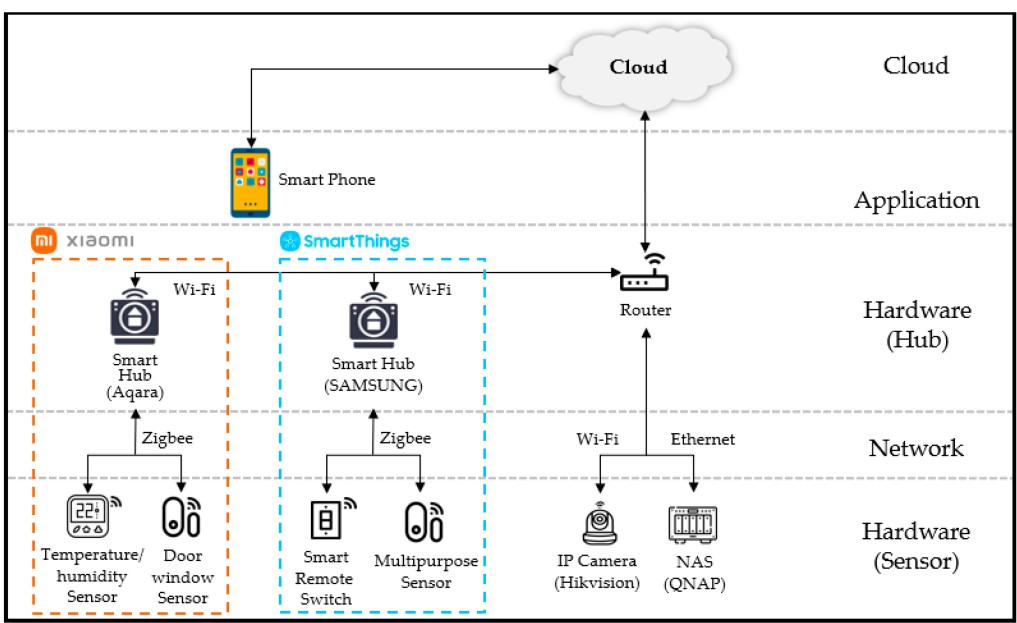

**Figure 7.** Experimental environment of a smart home.

**Table 3.** Devices employed in the experimental environment.

| Manufacturer | Device | Model | Communication | Hardware Interface |
|---|---|---|---|---|
| Samsung | Smart Phone | Galaxy A12 | Wi-Fi | - |
| | Smart hub | IOT-V3P03 | Zigbee, Z-wave, Wi-Fi | O |
| | Multipurpose sensor | IOT-MPP03 | Zigbee | - |
| | Smart remote switch | WS-KRC01 | Zigbee | X |
| Aqara | Smart hub | HM1S-G01 | Zigbee, Wi-Fi | O |
| | Door window sensor | MCCGQ11LM | Zigbee | X |
| | Temperature/humidity sensor | WSDCGQ11LM | Zigbee | X |
| QNAP | NAS | TS-251D | Ethernet | O |
| Hikvision | IP Camera | DS-2CV2U21FD-IW | Wi-Fi | O |

**Table 4.** Smart-home applications and analysis environments.

| Item | Name | Version | Usage |
|---|---|---|---|
| Application | SmartThings | 1.8.01.22 | Monitoring and controlling smart-home devices |
| | Mi home | 8.6.710.2101 | |
| Cloud data acquisition | Postman | 9.8.13 | Cloud API call tool |
| Network packet data analysis | Wireshark | 4.2.0 | Network protocol analyzer |
| Hardware interface analysis | PuTTy | 0.79 | Image dump in serial communication |
| Application internal storage analysis | FTK Imager | 4.7.1.2 | Data structure analysis of application storage |
| | DB Browser for SQLite | 3.12.2 | Database analysis on Application |

*4.1. OSINT Data Acquisition*

In this section, the OSINT-based data acquisition process, as detailed in Section 3, is applied to the systematic gathering and analysis of publicly available data from SmartThings and Aqara Hubs, QNAP NAS, and Hikvision IP Cameras.

### 4.1.1. SmartThings Hub

Initially, publicly available information about SmartThings devices, such as basic specifications, was collected from the official website and through the federal communications commission (FCC) ID. This information included the model code, Ethernet port, Wi-Fi, supported wireless networks, and input voltage. A subsequent investigation into the vulnerabilities of the SmartThings hub revealed vulnerabilities in previous generations, but no registered common vulnerabilities and exposures (CVEs) were found for this device.

Moreover, 132 publicly available APIs related to the SmartThings hub were identified, and an additional API was discovered through a static analysis of the HTML on the official website. Most of these 133 APIs provide information relevant to an incident investigation, such as device configuration information and event logs. Table 5 categorizes all APIs according to their contribution to an incident investigation.

**Table 5.** Level of attribution of the acquired application programming interface (API).

| Category | Count | Description |
|----------|-------|-------------|
| Essential | 16 | APIs that directly provide device status and related information |
| Considerable | 12 | APIs that cannot directly provide incident-related information but can provide additional information about the device |
| Irrelevant | 105 | APIs that cannot directly provide incident-related information but can provide additional information about the device |

### 4.1.2. Aqara Hub

Initially, publicly available information about Aqara devices was collected from the Aqara website and through the FCC ID, including the model name, input voltage, input current, power, supported wireless protocols, operating temperature, and humidity. Following this, an investigation into the vulnerabilities of the Aqara hub was conducted, but no registered CVEs were found. However, various documents were obtained indicating the possibility of exploiting the device using miiocli, an open-source tool for smart-home control.

Only five publicly available APIs related to Aqara were identified, and all of these APIs engaged in functions related to personal information, such as user account details. The APIs providing access to information (e.g., device event logs) were not public. Nevertheless, open-source tools using APIs that can acquire device information were discovered. By analyzing the code of these tools, an additional two APIs were identified.

### 4.1.3. QNAP NAS

In this research, publicly accessible data targeting the QNAP NAS devices were collected. Initially, fundamental details about the QNAP NAS devices, such as charging standards, external interfaces (e.g., HDMI ports), and intricate specifics of the CPU architecture and chip, were obtained from the manufacturer's documentation. During the vulnerability assessment of the QNAP NAS devices for this experiment, nine CVE entries were identified. These data (gathered via OSINT methodologies) and the CVE database were instrumental when executing exploits to extract information from the application, network, and hardware perspectives.

Significantly, QNAP NAS devices do not have an associated cloud service; thus, they lacked publicly available APIs for cloud data access. Furthermore, the absence of unofficial API documentation precluded the use of API-based data collection methods. Consequently, this necessitated a shift to alternative investigative processes for data acquisition.

### 4.1.4. Hikvision IP Camera

We collected publicly available data related to Hikvision cameras. The manufacturer provided us with information that allowed us to verify the wireless local area network standards, supported protocols, and functionalities of these specific devices. Addition-

ally, during our investigation of the vulnerabilities of Hikvision cameras, we successfully gathered details on four distinct CVEs.

While the Hikvision IP cameras could be managed through an application, and it was presumed that an API existed, we were unable to find any publicly accessible API documentation or unofficially used APIs. As a result, we concluded that data acquisition through the API was impractical. This situation required the adoption of alternative investigative procedures for further data collection.

### 4.2. Application Data Acquisition

In this section, the application-based data acquisition process from Section 3 was applied to SmartThings and Aqara hubs. It details the use of FTK Imager and SQLite for analyzing SmartThings application data and similar methods for Aqara hub's Mi Home application. This section focuses on extracting and analyzing application-level data, crucial for understanding security in IoT devices.

#### 4.2.1. SmartThings Hub

The SmartThings hub and sensors operate within the SmartThings platform and are controlled via the SmartThings app. Information from the sensors and that related to the hub is conveyed to users through the app and is stored in the application's internal storage. These data can be identified and collected via a rooted smartphone. The package folder of the SmartThings app is in the internal storage at /data/data/com.samsung.android.oneconnect.

Data identification was conducted using FTK Imager, and the SmartThings platform employs an SQL database for storing various logs and information. The analysis of the database files was conducted using SQLite. There were 60 database files, encompassing 287 tables and 2014 columns. Among these numerous database files, limited data were identifiable and utilizable for incident investigation. For an effective categorization of data pertinent to the investigation, the databases were divided into essential, considerable, and irrelevant. Accordingly, four databases were categorized as essential, 23 as considerable, and 33 as irrelevant for the investigation. The essential category included information such as the list of devices connected to the SmartThings platform, sensor event logs, and smart-home creation events. The considerable category encompassed user account information, Wi-Fi MAC addresses, OS information, and device IP information, which were deemed valid for the investigation of security breaches. Figure 8 presents history.db, one of the required DBs where sensor logs are stored.

| | text | activityType | activityPayload | messageTime | epoch | hash | uiTimestamp |
|---|---|---|---|---|---|---|---|
| | Filter | Filter | Filter | Filter | Filter | Filter | Filter |
| 1 | accelerationSensor: inactive | DEVICE | {"attributeName":"acceleration","attributeValue":... | 1684673855090 | 1684673855090 | 2672474870 | 1684676444120 |
| 2 | temperatureMeasurement: 23.6 °C | DEVICE | {"attributeName":"temperature","attributeValue":... | 1684673946457 | 1684673946457 | 2222705421 | 1684676444120 |
| 3 | soundSensor: detected | DEVICE | {"attributeName":"sound","attributeValue":"detec... | 1684674004526 | 1684674004526 | 2833721336 | 1684676444120 |
| 4 | soundSensor: not detected | DEVICE | {"attributeName":"sound","attributeValue":"not ... | 1684674019950 | 1684674019950 | 135596340 | 1684676444120 |
| 5 | soundSensor: detected | DEVICE | {"attributeName":"sound","attributeValue":"detec... | 1684674022289 | 1684674022289 | 1526769265 | 1684676444120 |
| 6 | rssi: -34.0 dBm | DEVICE | {"attributeName":"rssi","attributeValue":"-34","ca... | 1684674026962 | 1684674026962 | 2430476536 | 1684676444120 |
| 7 | lqi: 100 | DEVICE | {"attributeName":"lqi","attributeValue":"100","ca... | 1684674026964 | 1684674026964 | 2222970570 | 1684676444120 |
| 8 | soundSensor: not detected | DEVICE | {"attributeName":"sound","attributeValue":"not ... | 1684674037823 | 1684674037823 | 3662983283 | 1684676444120 |
| 9 | soundSensor: detected | DEVICE | {"attributeName":"sound","attributeValue":"detec... | 1684674076677 | 1684674076677 | 3574924938 | 1684676444120 |
| 10 | temperatureMeasurement: 23.4 °C | DEVICE | {"attributeName":"temperature","attributeValue":... | 1684674078981 | 1684674078981 | 1050444305 | 1684676444120 |

**Figure 8.** Internal sensor logs in history.db.

#### 4.2.2. Aqara Hub

In this experiment, the Aqara hub and associated sensors, functioning within the Xiaomi Smart-Home platform, were controlled via the Mi Home application. During data transmission to users, the Aqara hub generates relevant artifacts within the application's

internal storage. These artifacts are stored in the package folder of the Mi Home app located at the path/data/data/com.xiaomi.smarthome/ and include information related to the smart home and logs from the sensors.

The package folder of the Aqara hub was extracted using a rooted smartphone, and data identification was performed using the FTK Imager. Within the Xiaomi Smart-Home platform, three database files and 104 XML files were identified. Data collection from the database files proved challenging due to obfuscation and identification difficulties. The XML files were categorized based on their relevance in the event of a security breach. Three files fell into the essential category, two into the considerable category, and 99 were deemed irrelevant. The essential category comprised log information from sensors and devices. The considerable category contained the device IP, unique model name, and initial launch time of the application. Finally, the irrelevant category included metadata unrelated to smart-home devices, instead pertaining to the smart-home app. Notably, the essential sensor and device logs were located within the package folder at files/plugin/install/rm/{Plugin name}/data/{Device ID}/data/config.xml. Figure 9 provides an example of the door open/close sensor logs generated in the experimental environment.

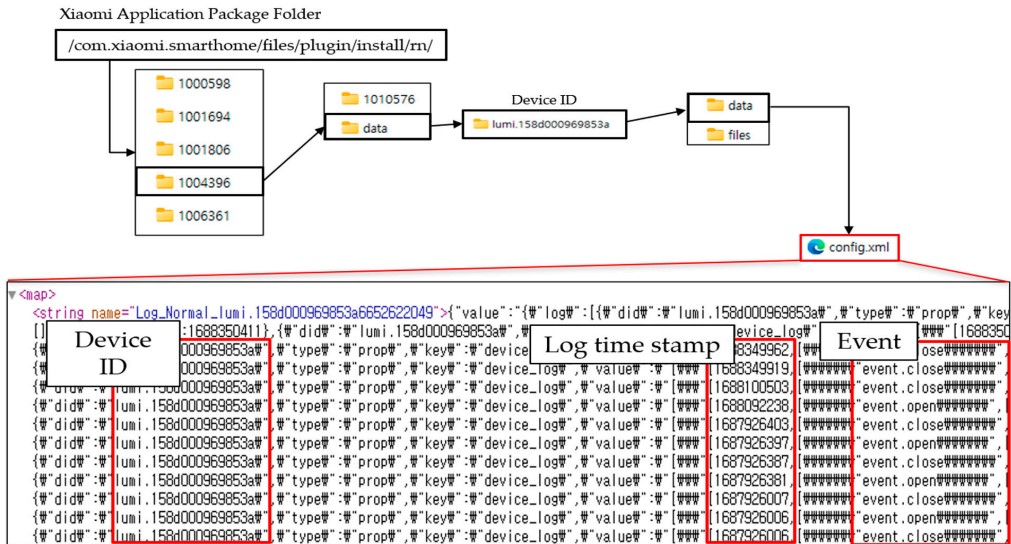

**Figure 9.** Sensor log information linked to the Aqara hub.

### 4.3. Network Data Acquisition

In this section, the network-based data acquisition process, as described in Section 3, is utilized to systematically gather and analyze network-related data from the Aqara Hub, QNAP NAS, and Hikvision IP Camera. This involves employing tools such as NMAP for port scanning and other methodologies to explore network services and identify vulnerabilities.

### 4.3.1. Aqara Hub

We identified open ports in the range from 1 to 65,535 using the NMAP tool. However, the port scan results revealed no open ports. Given the smart-home characteristic of communicating with an external network, such as the cloud, this can be interpreted as a security measure to hide open ports.

The tools and exploits collected through OSINT were used to access the device remotely. A vulnerability due to inadequate authentication validation in miiocli, an open-source tool for smart-home control, was exploited to create account information and open a telnet port, subsequently activating the telnet service. This approach allowed for remote access and entry into the ash shell.

The smart-home hub communicates with devices using the low-power network protocol Zigbee; thus, processes handling communication between these devices must be

constantly active. Therefore, upon analyzing the running processes, we were able to identify a process managing the Zigbee network logs, as illustrated in Figure 10.

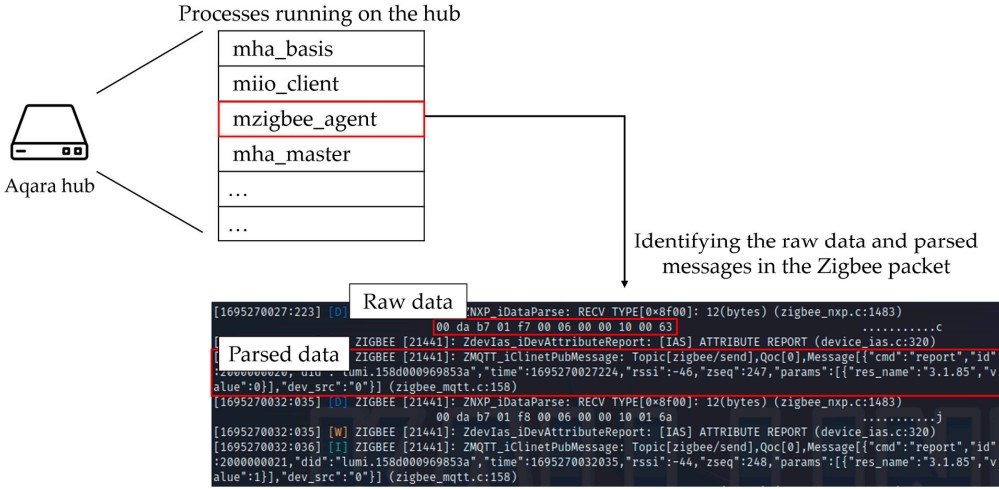

**Figure 10.** Zigbee logs from Aqara hub.

### 4.3.2. QNAP NAS

We confirmed that various ports were open by scanning the open ports on the activated NAS using the NMAP tool. Among the open ports, ports 22 and 23 were the ports for OpenSSH and telnet, respectively, indicating that remote access to the device was possible. In addition to the ports enabling remote access, an HTTP port could be identified that allowed for access to the web interface of the NAS service. When accessing the device's IP through a web browser, a web interface providing equipment configuration information and management features was observed. The information that could be obtained from the web interface included the following:

- System information
- Network status
- System services
- Hardware information
- Resource monitor
- Remote service information

Port 22 was identified; thus, SSH communication with the NAS was established using a serial communication tool, such as PuTTy. The terminal had root privileges, allowing for direct access to major files and data in the system without additional permission requests. The root directory contained 19 directories and 4 files, similar to a typical Linux system.

The subsequent data extraction and analysis revealed that 21 files included the characteristic and critical system information of the NAS device. Table 6 lists the files that must be checked to obtain the system information from the NAS device.

**Table 6.** Critical information obtainable from the NAS equipment.

| File Name | Main Content |
|---|---|
| /proc/cpuinfo | Processor information |
| /proc/devices | List of device drivers configured in the currently running kernel |
| /proc/diskstats | I/O statistics for block devices |
| /proc/filesystems | Filesystems supported by the NAS device |
| /proc/locks | Kernel lock information |
| /proc/mdstat | Information about RAID |
| /proc/meminfo | Memory usage |
| /proc/mounts | Mounted file systems |

**Table 6.** *Cont.*

| File Name | Main Content |
|---|---|
| /proc/partitions | Partition table known to the system |
| /proc/slabinfo | Memory usage at the slab level |
| /proc/stat | Overall statistics about the system |
| /proc/swaps | Information about swap space usage |
| /proc/uptime | System uptime |
| /proc/version | Linux and kernel version |
| /etc/group | User groups and their users |
| /etc/hosts | Host information |
| /etc/passwd | Account information |
| /etc/services | Information about supported services |
| /etc/resolv.conf | DNS settings |
| /etc/config/crontab | Automatic execution settings |
| /etc/config/raid.conf | Information about RAID |

### 4.3.3. Hikvision IP Camera

A scan was conducted on an IP camera using the NMAP tool to detect open ports and identify active services. This scan revealed that Port 22, commonly used for secure shell (SSH) communications, was open. It was possible to access a protected shell, commonly known as a busybox, through an established SSH connection.

However, the capabilities of the protected shell for acquiring information were limited, presenting a significant obstacle to data collection procedures. Additionally, busybox (particularly in embedded device environments) can vary in structure, leading to substantial differences in the supported commands, depending on the device. For the Hikvision IP camera, 101 commands were supported, among which no commands for establishing arbitrary connections with external networks were identified.

### 4.4. Hardware Data Acquisition

In this section, the hardware data acquisition process, as outlined in Section 3, was applied to the Hikvision IP Camera. The methodology encompassed disassembling the camera for a detailed analysis of key hardware components, including CPU and flash memory, and employed techniques such as memory dumping for comprehensive data extraction.

Hikvision IP Camera

In the process of examining the internal structure of the Hikvision cameras, removing the camera case allowed for the clear identification of various critical hardware components. Figure 11 depicts the internal components of an IP camera. These include the CPU, flash memory chip, random access memory, and a debugging interface known as a universal asynchronous receiver transmitter (UART). It is worth noting that with Hikvision cameras, access through UART and the use of a "magic key" enabled entry into the U-boot shell. This U-boot shell provided commands that allowed for the extraction of crucial hardware debugging information, including detailed memory and CPU data. However, access to the JTAG debugging interface was challenging and remained unexplored.

Additionally, by leveraging an OSINT approach, the datasheet for the Hikvision camera flash memory was acquired. This approach allowed for the precise identification of the memory pin configuration. After this identification, a memory dump process was executed, and the acquired binary files were carved using the Binwalk tool. This procedure successfully extracted 3836 directories and files, including 18 directories commonly found in Linux systems. Among the retrieved information were details of accounts and hosts commonly observed in Linux systems. This data provided insight into the types of data that could be accessed and collected through the hardware collection process.

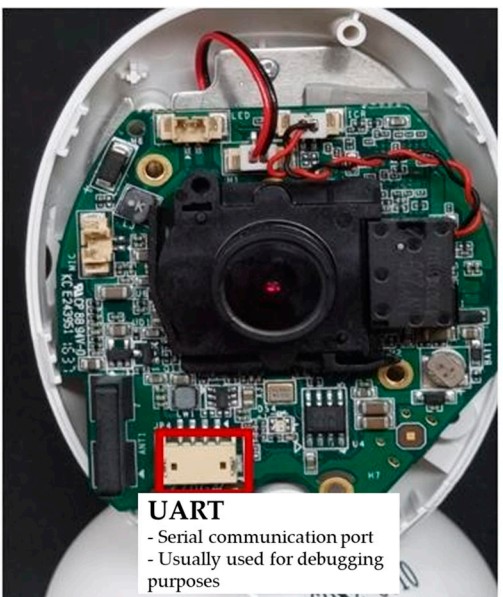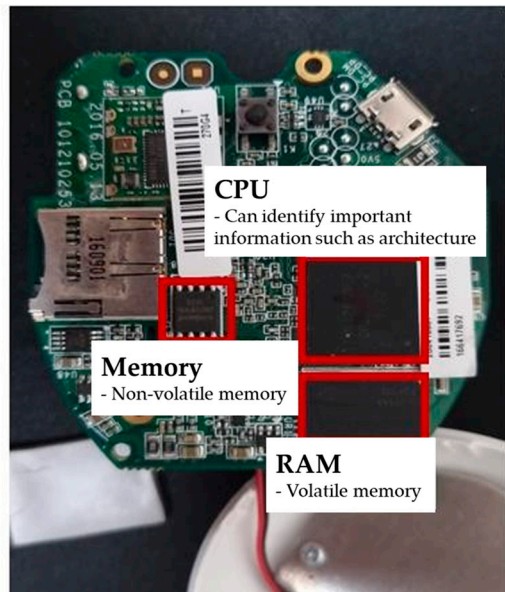

**Figure 11.** Hikvision IP camera hardware components.

## 5. Incident Investigation in Smart Homes

This study considers the heterogeneity of IoT devices and includes diverse types in the experiments. Additionally, we established both a cloud-dependent IoT ecosystem and an independent IoT ecosystem that does not rely on cloud services. This approach focuses on reproducing intrusion scenarios that can occur in actual environments.

There are two main types of scenarios: botnet malware and intrusion. These represent common types of cybercrime in actual IoT environments. Reproducing realistic threats and collecting data helps validate the effectiveness of the proposed methodology.

### 5.1. Scenario 1: Botnet Malware

This scenario was selected for an intrusion incident experiment based on the Mirai malware. Historically, Mirai malware has posed a significant threat to architecture, leading to ongoing concerns. Given this background, the risks posed by Mirai malware and its potential applicability to different system architectures must be examined.

The Mirai malware samples used in this experiment were collected from a publicly available malware sample database. The devices targeted for the experiment were the QNAP NAS and Aqara hub, chosen because of their known vulnerabilities and attack vectors that allowed for internal system access, making them suitable for Mirai malware infection. This method facilitated a better understanding and analysis of the malware activity patterns.

The malware injection in this scenario was conducted through telnet access to the target device system, where the malware was directly downloaded and executed. Figure 12 illustrates the scenario setup. This scenario assumed that the malware infection occurred due to exposure from another device, and the experiment was conducted accordingly. The stages of data collection in this scenario were as follows.

- Execute malware
- Scan port for the IP address
- Device system access through a dictionary

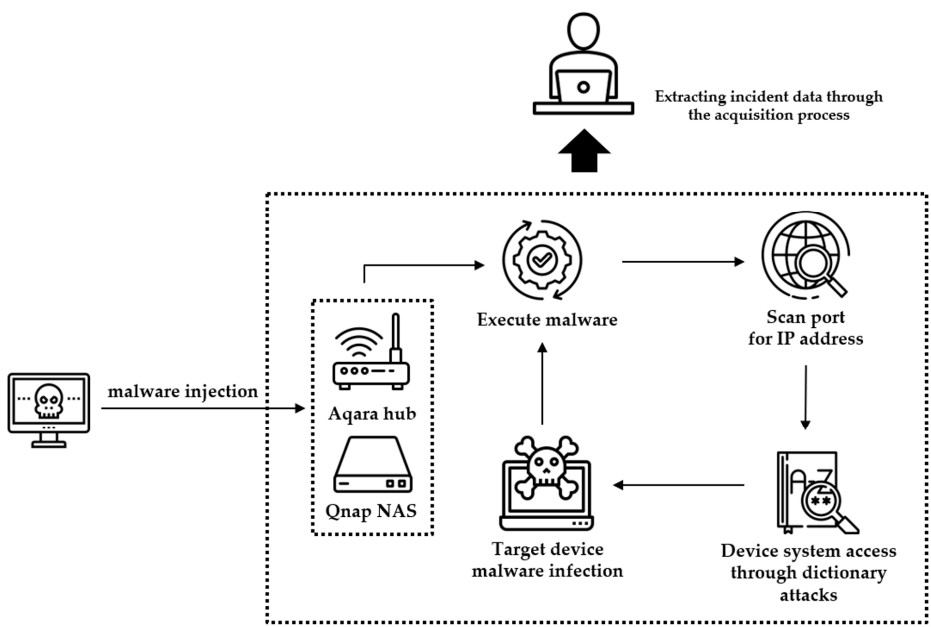

**Figure 12.** Botnet scenario diagram.

5.1.1. Incident Investigation: NAS

A port scan was conducted on the NAS. During this process, we discovered that telnet and NAS web service ports were both open, allowing us to identify malware artifacts and confirm the presence of an infection. When malware is executed on the NAS, it can be detected through remote access or web services. The web service interface enables the inspection of remote login records, identifying unauthorized IP access, which aids in identifying security breaches. Moreover, remote access to the NAS system reveals historical information, process data, and login records. The execution of malware is further confirmed by identifying abnormal strings in commands or processes designed for initiating the malware.

The attempt to infect external target devices was evidenced through port scanning activities, detectable via packet sniffing. Activities involving sending synchronize packets to telnet Port 22 of random target IPs to check for open ports were observed. Additionally, the attempt to access external target devices using predefined ID/password combinations was also identified.

5.1.2. Incident Investigation: Hub

A port scanning operation was conducted on Aqara devices. During this process, all ports on the Aqara hub were closed. However, a flaw in Xiaomi's control tool for its smart-home platform (gathered using OSINT) was exploited to open a telnet port on the Aqara hub and establish remote access. This approach allowed for the identification and verification of malware artifacts present in the Aqara hub. The Aqara hub does not maintain a history file like standard Linux systems, making it impossible to collect and confirm commands executed by the malware. Nevertheless, we examined the process information and identified obfuscated processes in operation, confirming active malware execution.

The attempt to infect external target devices followed a method similar to that used with the previously mentioned NAS devices. Aqara devices were subjected to packet sniffing, which facilitated the identification of malicious activities and botnet infections, similar to the NAS. These investigative efforts highlight the effectiveness of a network-centric approach in identifying and analyzing botnet artifacts when a system has been compromised by a botnet.

*5.2. Scenario 2: Intrusion*

This scenario assumes a situation in which an external attacker has infiltrated an IoT environment and taken control of the system by means other than malware. Because of their limited resources, IoT devices have relatively fewer means to defend against external attacks compared with other network devices. Considering this, collecting artifacts of attacks initiated by an intruder in IoT devices is a critical task.

In this scenario, the SmartThings hub was selected as the target of the attack. The SmartThings hub stores most information in the cloud and on mobile devices; thus, the investigation process involved collecting data through OSINT and application aspect processes. This approach was taken to explore the possibility of identifying abnormal activities by the external intruder. Figure 13 presents the types of external attack scenarios used to validate the methods of data collection.

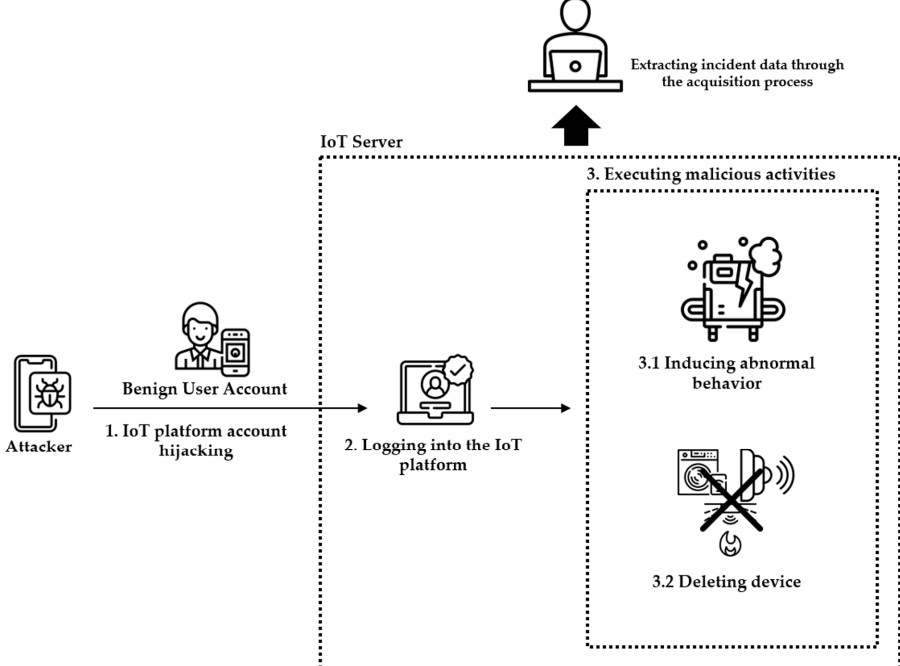

**Figure 13.** Intrusion scenario diagram.

### 5.2.1. Inducing Abnormal Behavior

This scenario was designed with the objective of disrupting the normal operation of a device. Frequent malfunctions can desensitize the device to the risk of actual malfunctions that may occur later. Therefore, such malicious actions interfere with the usability of the device and lead to a decline in user trust in their devices.

Event logs of the device actions are recorded in the cloud. However, directly accessing the cloud to collect artifacts is typically not permitted. Therefore, APIs must be used to obtain some data stored in the cloud, which are part of the OSINT of incident investigation. The data acquisition from the OSINT perspective was conducted accordingly.

The API URL related to the device's event log is /history/devices. While this is not a public API, it can be obtained through static analysis of the web interface designed for platform management. Investigators can use the event logs obtained through this API to cross-verify data with the user statements to determine information, such as which actions were not intended by the user.

### 5.2.2. Deleting Device

This scenario involves an external attacker deleting a device from the interconnected platform, aimed to reduce user convenience and leave the device vulnerable. The deletion of a device does not leave direct logs in the cloud. However, when a device is deleted, this

information is typically synchronized with the associated mobile application. Therefore, information about the device deletion can be obtained through incident investigation on the application side.

Changes occur in the application layout when a device is added or deleted. The PersistentLogData.db file, located in the internal storage of the mobile device, manages the user-interface changes in the SmartThings application. This file includes a timestamp that indicates when the changes were made. By analyzing this file, the following deletion information can be obtained:

- Deleted device ID
- Deleted device name
- Device installation location information
- Information about the hub connected to the device

## 6. Discussion

This section provides a detailed analysis of the results and insights obtained from our experimental setup, focusing on data acquisition from the SmartThings and Aqara platforms, as well as from the NAS and IP camera devices. Our experiments with the SmartThings and Aqara platforms demonstrated that effective data collection is feasible through OSINT and applications, regardless of the device's operational state. For active devices, network-based data acquisition methods proved efficient. However, in inactive states, we shifted to hardware-based approaches. This two-pronged strategy allowed for versatile and adaptable data collection.

Data gathered from NAS devices in active states included network configurations, service statuses, and process-related information. Conversely, when these devices were inactive, we could obtain file system data, details on malicious files, and log data. A significant challenge was hardware disassembly, especially as UART and JTAG connections required soldering for access, as they were part of the printed circuit board.

Regarding IP camera devices, active data acquisition provided access to network details, account information, and service configurations. Remote services like SSH could be utilized for additional data collection. Disassembling the hardware, we achieved system access via UART ports, although limited to a restricted shell rather than a full root shell. Despite this limitation, it allowed us to gather crucial network and system information. Additionally, memory dumping was feasible using a small outline integrated circuit test clip on the accessible 8-pin flash memory.

We encountered several challenges in our research:

- Disparities in API requirements across different platforms necessitated varied methodologies and scopes for data acquisition. Relying on APIs for OSINT often required specific account information and did not encompass all cloud-stored data, leading to gaps in comprehensive data collection.
- Accessing hardware debugging ports was complicated due to security measures implemented by manufacturers, posing a challenge in physically examining the devices.
- The encryption of network packet data, particularly the use of TLS encryption by many smart-home devices, made intercepting data transmissions difficult. Theoretical methods like TLS downgrade attacks were conceivable, but their practical application was limited due to the need to modify smart-hub root certificates, hindering effective packet decryption.

In response to these challenges, we explored alternative methods, such as using encrypted network packet headers to identify device manufacturers or types. Although this did not grant direct access to communication content, it provided valuable insights into the types of operational devices.

In the field of OSINT investigations, retrieving personal data from cloud-based platforms is a key focus area, presenting a significant challenge due to limited access. This highlights a major constraint in cloud-based digital forensics. Investigations involving

application data are also hampered by a reliance on mobile devices, potentially leading to gaps or inaccuracies in the gathered information. Network data acquisition, primarily focused on intercepted data, faces hurdles due to widespread encryption, complicating the analysis of communication content. In terms of hardware investigations, both chip-off analysis and debugging port analysis have their inherent limitations: chip-off analysis risks irreversible device damage and accessing debugging ports is typically successful only with less secure devices [34]. These challenges emphasize the need for developing diverse and adaptable forensic methodologies within IoT environments, considering the varying security measures and data accessibility issues across different devices and platforms.

These findings underscore the complex nature of digital forensics in smart-home environments. The variety of devices and platforms calls for a dynamic and adaptive approach to data acquisition. Understanding and tackling these challenges is essential for advancing forensic methodologies in this rapidly evolving field [35–37].

## 7. Conclusions

This research has explored the complexities of conducting digital forensics in heterogeneous IoT environments, specifically focusing on smart-home technology. We developed a comprehensive methodology for data acquisition and analysis, which emphasizes four key aspects: OSINT, application, network, and hardware. Our examination of various IoT devices and platforms, such as Samsung SmartThings, Aqara, QNAP NAS, and Hikvision IP cameras, has provided insights into their vulnerabilities and potential forensic artifacts.

Our findings highlight the importance of a multifaceted and adaptive approach for effective forensic analysis in smart-home environments. We established that critical forensic data can be acquired through different methods, depending on the operational state of IoT devices. Network-based data acquisition is effective for active devices, while hardware analysis is more beneficial for inactive devices. The study has also revealed several challenges, including variations in API requirements across platforms, complexities in accessing hardware debugging ports, and decrypting encrypted network packet data. These challenges caused us to adapt our methodologies and employ alternative strategies, such as analyzing encrypted network packet headers.

Looking ahead, this research lays the foundation for future advancements in IoT digital forensics. One of the key areas of focus will be expanding forensic analysis to encompass a wider range of IoT devices and scenarios. This expansion will involve refining data collection methodologies to handle encrypted network data and complex hardware interfaces more effectively. Special attention will be given to analyzing encrypted network packet headers, with the aim of extracting more detailed information, such as the type and manufacturer of devices. This enhanced approach is expected to provide a deeper understanding of the intricate encrypted traffic generated by IoT devices and facilitate the extraction of more detailed data. These improvements are anticipated to significantly enhance the robustness and comprehensiveness of digital forensic investigations in smart-home environments.

Additionally, future research will also concentrate on strengthening forensic processes related to the physical memory and hardware aspects of heterogeneous IoT devices. This initiative is expected to significantly improve the efficacy of digital forensics in smart-home settings, opening up new avenues for investigation and strengthening the security infrastructure of these increasingly prevalent environments. By advancing these areas, our goal is to keep pace with, and stay ahead of, the evolving cybersecurity threats in smart-home ecosystems, ensuring a safer digital future.

**Author Contributions:** Conceptualization, D.-H.S. and S.-J.H.; methodology, D.-H.S. and Y.-B.K.; validation, S.-J.H. and Y.-B.K.; investigation, S.-J.H. and Y.-B.K.; writing—original draft preparation, D.-H.S., S.-J.H. and Y.-B.K.; writing—review and editing, D.-H.S. and I.-C.E.; project administration, I.-C.E.; funding acquisition, I.-C.E. All authors have read and agreed to the published version of the manuscript.

**Funding:** This work was supported by the Institute for Information & communications Technology Planning & Evaluation (IITP) grant funded by the Korea government (MSIT) (No.2022-0-01203, Regional strategic Industry convergence security core talent training business). Additionally, the results of this study were also supported by the Nuclear Safety Research Program through the Korea Foundation of Nuclear Safety (KoFONS), using the financial resource granted by the Nuclear Safety and Security Commission (NSSC) of the Republic of Korea (No.2106061).

**Institutional Review Board Statement:** Not applicable.

**Informed Consent Statement:** Not applicable.

**Data Availability Statement:** The data used for this research are contained or presented in the tables in the article.

**Conflicts of Interest:** The authors declare no conflicts of interest.

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
