# Peer review of "Research on Digital Forensics Analyzing Heterogeneous Internet of Things Incident Investigations"

_applsci, doi:10.3390/app14031128_

Round 1
Reviewer 1 Report
Comments and Suggestions for Authors
The paper explores the challenges of digital forensics within the Fourth Industrial Revolution, specifically focusing on the integration of the Internet of Things (IoT) into smart-home technology. The overall structure is well organized, well-written, and easy to comprehend. The authors consistently employ a robust methodology, incorporating open-source intelligence, application, network, and hardware analyses. The inclusion of practical experiments on platforms like SmartThings, Aqara, QNAP NAS, and Hikvision IP cameras enhances the study's real-world relevance.
However, in my opinion, some aspects need improvement:
-
- 1. The English throughout the paper requires a revision due to numerous typos.
-
- 2. Some figures, such as 9 and 10, lack clarity in their presentation, especially regarding the purpose of displaying logs. Figure 11, showing Hikvision IP camera hardware components, could benefit from an explanation of its significance. Furthermore, Figure 8, featuring text in another language, needs translation.
-
- 3. In section 5, incidents related to security are investigated, but the introduction and related work sections lack discussions on the types of attacks and proposed software solutions. Addressing this gap would enhance the paper's comprehensiveness.
4. Additionally, it would be beneficial for the paper to discuss limitations explicitly. For instance, addressing the generalizability of the findings to a broader range of IoT devices and platforms would enhance the paper's transparency.
- 3. In section 5, incidents related to security are investigated, but the introduction and related work sections lack discussions on the types of attacks and proposed software solutions. Addressing this gap would enhance the paper's comprehensiveness.
-
- 4. Finally, the conclusion lacks perspective. Indeed, it would be beneficial to propose future works including suggestions for further research and development in the field of IoT digital forensics will provide a more comprehensive conclusion.
In summary, the paper is an interesting read, making a valuable contribution to the field of IoT digital forensics. It effectively combines theoretical insights with practical experimentation, addressing the multifaceted challenges associated with smart-home environments.
Comments on the Quality of English LanguageEnglish is globally well writen. Some typos are detected and need to be corrected in the revised version
Reviewer 2 Report
Comments and Suggestions for Authors
This manuscript applsci-2813168 proposed focusing on developing forensic methodologies suitable for the diverse and complex world of smart-home IoT devices. This research is contextualized within the rising trend of interconnected smart homes and their associated cybersecurity vulnerabilities. Methodologically, the authors formulate a comprehensive approach combining open-source intelligence, application, network, and hardware analyses, aiming to accommodate the operational and data storage characteristics of various IoT devices. Extensive experiments were conducted on prevalent platforms, such as Samsung SmartThings, Aqara, QNAP NAS, and Hikvision IP cameras, to validate the proposed methodology. These experiments revealed crucial insight into the complexities of forensic data acquisition in smart-home environments, emphasizing the need for customized forensic strategies tailored to the specific attributes of various IoT devices. The study significantly advances the field of IoT digital forensics and provides a foundational framework for future explorations into broader IoT scenarios. It underscores the need for evolving forensic methodologies to keep pace with rapid technological advancements in IoT. It was a pleasure reviewing this work and I can recommend it for publication in Applied Science after a major revision. I respectfully refer the authors to my comments below.
1. The English needs to be revised throughout. The authors should pay attention to the spelling and grammar throughout this work. I would only respectfully recommend that the authors perform this revision or seek the help of someone who can aid the authors.
2. (Page 1, Section 1. Introduction) The reviewer suggests authors don't list a lot of related tasks directly. It is better to select some representative and related literature or models to introduce with certain logic. For example, the latter model is an improvement on one aspect of the former model.
3. Experimental pictures or tables should be described and the results should be analyzed in the picture description so that readers can clearly know the meaning without looking at the body. For example, in Figures 9-13, the results should be marked by different arrow to show the advantage of the proposed method. The author can describe the results of the analysis of this phenomenon.
4. The authors are suggested to add some experiments with the methods proposed in other literatures, then compare these results with yours, rather than just comparing the methods proposed by yourself on different models.
5. (Page 2, Section I Introduction) Please add some references. The original statement is suggested as “bilities like artificial intelligence (AI) and big data [1]” ([1] "Orientation Cues-Aware Facial Relationship Representation for Head Pose Estimation via Transformer," IEEE Transactions on Image Processing, vol. 32, pp. 6289-6302, 2023.)
6. Discuss the pros and cons of the proposed model.
7. An introductory sentence should be given at the beginning of each section.
8. The authors are suggested to add some experiments with the methods proposed in other literatures, then compare these results with yours, rather than just comparing the methods proposed by yourself on different models.
My overall impression of this manuscript is that it is in general well-organized. The work seems interesting and the technical contributions are solid. I would like to check the revised manuscript again.
Comments on the Quality of English LanguageThe English needs to be revised throughout. The authors should pay attention to the spelling and grammar throughout this work. I would only respectfully recommend that the authors perform this revision or seek the help of someone who can aid the authors.
